# The Role of Elements Partition and Austenite Grain Size in the Ferrite-Bainite Banding Formation during Hot Rolling

**DOI:** 10.3390/ma14092356

**Published:** 2021-05-01

**Authors:** Yina Zhao, Yinli Chen, He Wei, Jiquan Sun, Wei Yu

**Affiliations:** 1Collaborative Innovation Center of Steel Technology, University of Science and Technology Beijing, Beijing 100083, China; b20190527@xs.ustb.edu.cn (Y.Z.); weihe@xs.ustb.edu.cn (H.W.); sunjq@ustb.edu.cn (J.S.); 2Institute of Engineering Technology, University of Science and Technology Beijing, Beijing 100083, China; yuwei@nercar.ustb.edu.cn

**Keywords:** partition, diffusion, DICTRA, matlab, phase transition equilibrium model, austenite grain size, ferrite/bainite banding

## Abstract

The partitioning and diffusion of solute elements in hot rolling and the effect of the partitioning and diffusion on the ferrite-bainite banding formation after hot rolling in the 20CrMnTi steel were experimentally examined by EPMA (electron probe microanalysis) technology and simulated by DICTRTA and MATLAB software. The austenite grain size related to the hot rolling process and the effect of austenite grain size on the ferrite-bainite banding formation were studied. The results show that experimental steel without banding has the most uniform hardness distribution, which is taken from the edge of the cast slab and 1/4 diameter position of the cast slab, heating at 1100 °C for 2 h and above 1200 °C for 2–4 h during the hot rolling, respectively. Cr, Mn, and Si diffuse and inhomogeneously concentrate in austenite during hot rolling, while C homogeneously concentrates in austenite. After the same hot rolling process, ΔA_e3_ increases and ferrite-bainite banding intensifies with increasing initial segregation width and segregation coefficient K of solute elements. Under the same initial segregation of solute elements, ΔA_e3_ drops and ferrite-bainite banding reduces with increasing heating temperature and extension heating time. When ΔA_e3_ drops below 14 °C, ferrite-bainite banding even disappears. What is more, the austenite grain size increases with increasing heating temperature and extension heating time. When the austenite grain size is above 21 μm, the experimental steel will not appear to have a banded structure after hot rolling.

## 1. Introduction

The emergence of banded structures such as bainite, martensite, pearlite, proeutectoid ferrite, and acicular ferrite banding in low carbon steel occupy a major position, and it has gradually aroused the attention of the majority of scientific researchers [1,2]. Gear steel is mostly low-carbon alloy steel, which is usually used to make larger wear-resistant parts in the machine. The banded structure in the steel is more serious, which increases the deformation during the carbon leakage process. The small diameter difference of each part of the height affects the fatigue life of the gear [3,4,5,6]. In the process of China’s leap from a major manufacturing country to a manufacturing power, high-grade machinery manufacturing has continuously increased the requirements for steel, and the banded structure is bound to be one of the issues affecting the quality stability and consistency of high-grade steel. However, foreign countries have been studying the banded structure since the early 1950s, but how to eliminate or effectively reduce its influence on the mechanical properties of steel has not yet been completely resolved. There are still very few domestic relevant studies. As the problem of banded structure’s influence on the production and service performance of steel is becoming more and more prominent, the problem of banded structure should be paid more attention to, especially in key areas (such as aerospace, marine engineering, advanced rail transit Equipment, etc.). The banded structure in the high-grade steel in service should receive urgent attention.

The research on banded structure can be traced back to the 1950s [7,8]. For Jatzchak, et al., the alloying element affects the transformation by affecting the carbon activity in the austenite during cooling and phase transformation, and Bastien et al. [9] believe that the alloying element changes the A_r3_ temperature, leading to proeutectoid ferrite shape Nuclear advances or suppresses these two views. Kirkaldy et al. [10] experimentally supported the theory of Bastien et al. and provided an empirical formula for the critical cooling rate as follows:$\;\dot{T}>5D\Delta T/\omega^{2}$, $\dot{\;T}$ is the critical cooling rate, D is the carbon diffusion coefficient, ΔT is the A_r3_ temperature difference caused by the alloying element, and ω is the width of the alloying element. At the end of the 20th century, GroBterlinden et al. [11] used a finite difference model to calculate the function of the austenite–ferrite interface change due to the cooling rate. Their model assumes that the decomposition of austenite is controlled by carbon diffusion. Thompson and Howell [12] considered the influence of the austenite grain size on the banded structure and showed that when the austenite grain size is greater than 2~3× the width of chemical segregation zone, the banded structure will not be formed. In 2002, Offerman et al. [13] quantified the nucleation and growth criteria for the transformation of isothermal austenite to ferrite. The results show that, in order to form a band structure, the relative difference between the ferrite nucleation rate in the region of the maximum and minimum transformation temperature should be greater than 6–8%. In 2016, Maalekian et al. [14] used the phase field method to simulate the effect of micro-segregation level, cooling rate, original austenite grain size, and banding width on the formation of banded structure with 2D periodic Mn band simulation. The simulation results are consistent with the experimental observations. As the cooling rate and austenite grain size decrease and the Mn segregation level increases, the banded structure becomes obvious.

Much work has been done to investigate how the microstructural banding is formed during hot rolling. Microstructural banding in low-alloy steel is due to the segregation of substitutional alloying elements during dendritic solidification [15,16,17]. The addition of elements such as manganese, chromium, and molybdenum cause solidification to occur over a range of temperatures and compositions. Consequently, the dendrite heart solidifies as relatively pure metal while the interdendritic spaces become enriched in solute. These high- and low-solute regions are elongated into parallel bands during rolling and forming operations [18,19]. Differences in austenite transformation behavior between bands lead to the formation of a laminated microstructure with discrete layers of martensite, bainite, ferrite, and pearlite [20]. The influence of the microsegregation of Mn, Si, and Cr on the austenite decomposition during isothermal transformations in hot-rolled medium carbon steel has been studied by neutron depolarisation, EPMA, and optical microscopy. Two-dimensional EPMA maps of the specimen annealed at 740 °C showed that microsegregation of alloying elements in hot rolled steel is strongly related to the ferrite/pearlite band formation. Similar EPMA maps for the specimen transformed at 680 °C demonstrate the presence of microchemical bands, while optical microscopy reveals the absence of microstructural bands. It is shown that the formation of microchemical bands is a prerequisite for band formation, but the kinetics of the phase transformation determine the actual formation of microstructural bands [21].

DICTRA is flexible software for the simulation of diffusion-controlled transformations in multicomponent alloys. It is closely linked with the Thermo-Calc software, which provides all necessary thermodynamic calculations [22,23]. DICTRA is a software package that simulates the diffusion control transition in a multi-element system. The DICTRA software can be used to simulate the element concentration distribution under different temperatures, times, and pressures [24]. Yao et al. [25] built the numerical model of HIP (Hot isostatic pressure) diffusion bonding, and distribution of elements and phases of DD407/FGH95 diffusion couples under different HIP temperature and bonding time were calculated with DICTRA and Thermal-Calc software. The simulated results indicated that compared with time, the temperature affects the diffusion of elements at the interface of diffusion bonding more. Hu Zhiliu et al. [26] analyzed the micro-segregation of Mg and Mn of Al-5.0 Mg-0.5 Mn alloy by DICTRA software. Mg-induced microsegregation can be eliminated by homogenization annealing at 470 °C for at least 11.1 h. However, Mn-induced micro-segregation cannot be eliminated by homogenization annealing in a short time.

Many scholars at home and abroad use model calculations to predict the structure performance. The binary system thermodynamic model includes the geometric repulsion model KRC model and the atomic interaction model LFG model, which are mostly used for the thermodynamic calculation of ordinary carbon steel. Aaronson, Domain, and Pound (ADP) et al., in order to study the phase transformation of high-strength steels with complex alloy systems, proposed a super-component model based on the KRC and LFG models, which did not consider the interaction between alloying elements. Peng [27] modified the two important parameters of Zener, considering the interaction of the alloying elements. Li et al. [28] considered the interaction between alloying elements to revise the super component thermodynamic model, and they used the revised model to calculate the influence of alloying elements and deformation energy storage in X100 pipeline steel on the ferrite phase transformation parameters. Phase transition thermodynamics determines the tendency of phase transition, and the calculation of phase transition equilibrium temperature A_e3_ of gear steel is necessary. Because of its simple mathematical expression and convenient application, the super-component model has been widely used in the study of phase transition thermodynamics of multi-element alloy systems. Wang et al. [29] used the super component model and the LFG model to calculate the phase transformation equilibrium temperatures A_e3_ and Ae 1 of TRIP steel, and found that they were in good agreement with the experimental values. Wang et al. [30] studied the isothermal transformation incubation period of austenite ferrite transformation. The actual ferrite transformation temperature A_r3_ of SS400 steel was calculated using Scheil’s additivity rule. The calculated A_r3_ is in good agreement with the measured A_r3_, which further demonstrates the accuracy of the phase transition incubation period model.

There has always been controversy about the research of banded structures. The use of observation methods and the continuous advancement of calculation and simulation to explain the banded structure from a more precise and in-depth perspective has just begun.

In this study, the distribution and diffusion of solute elements in 20CrMnTi steel during the hot rolling process were experimentally studied and simulated, and its influence on the formation of ferrite-bainite banded structure was analyzed. The austenite grain size related to the hot rolling process was counted, and the influence of the austenite grain size on the formation of ferrite-bainite banded structure was studied. The hardness along the normal direction of the experimental steel after hot rolling was detected, and the influence of the ferrite-bainite banded structure on the hardness distribution was analyzed. Therefore, by obtaining the best structure distribution, the mechanical properties of the sample steel are optimized.

## 2. Materials and Methods

The chemical composition (mass fraction, %) of the round cast slab is shown in Table 1. The section diameter of the round cast slab is 650 mm. The sampling method for observing the low power structure of circular billet is shown in Figure 1a. Firstly, hot hydrochloric acid was used to corrode the circular billet with an arc section of 325 mm × 75 mm × 20 mm, so as to observe its low-power dendrite morphology and take low-power photos. Then samples of 20 mm × 20 mm × 10 mm were taken from the heart, 1/4 diameter position, and edge along the radial direction of the cast slab. The dendrite test method proposed by predecessors [31,32] was adopted to conduct annealing treatment on the samples. The annealing process is shown in Figure 1b. First of all, the samples were kept warm at 930 °C for 30 min in a muffle furnace (TNX1400, Shanghai Xiangbei, Shanghai, China), then the muffle furnace was cooled to 680 °C, the temperature was kept warm for 2 h, and the samples were cooled to room temperature with the muffle furnace. After the heat treatment, the specimen was firstly milled off for 2 mm to avoid decarburization on the observation surface, and then polished. Four percent alcohol nitrate solution was used for erosion to prepare the metallographic specimen. The dendrite morphology was observed by LSCM (OLS4100 Laser scanning confocal microscope, OLYMPUS, Tokyo, Japan). In the EPMA (JXA-8230, JEOL, Tokyo, Japan) measurements; area analysis mode was used to obtain the C, Cr, Mn, Si, and Ti contents across regions of interest. The content of solute elements in ferrite and pearlite is measured by EPMA point-analysis mode.

Samples of 20 mm × 20 mm × 80 mm were taken from the center, 1/4 diameter position and edge of the casting slab, and their position is shown in Figure 1a; the rolling process is shown in Figure 1c; and the rolling schematic diagram is shown in Figure 1d. During the homogenization treatment, samples taken from the heart and edge of the cast slab were heated at 1100 °C for 2 h, and samples taken from 1/4 diameter position of the cast slab were heated at 1100 °C and 1150 °C for 2 h, and 1200 °C for 2 h and 4 h, respectively. After homogenization treatment, the materials were hot-rolled from 20 mm to 8 mm in thickness through the hot rolling mill (350 reversible), with a 60% thickness reduction. After rolling, these samples were cooled in the air and in the water respectively. The observed surface of them was parallel to the direction of rolling deformation, which is shown in Figure 1c. After polishing, 4% nitrate alcohol was used to erode the sample cooled in air, and the microstructure was observed by LSCM and SEM (scanning electron microscope). Saturated picric acid solution at 65 °C was used to boil samples cooled in water for 2 min, and the microstructure of austenite grain was observed by LSCM. In addition, in order to study the formation of the banded structure, the contents of solute elements in banded structure were detected by EPMA.

In this study, the partitioning and diffusion of solute elements in hot rolling and the effect of the partitioning and diffusion on the ferrite-bainite banding formation after hot rolling in the 20CrMnTi steel were experimentally examined by EPMA technology and simulated by DICTRTA and MATLAB software. The austenite grain size related to the hot rolling process and the effect of austenite grain size on the ferrite-bainite banding formation were studied. The hardness along the normal direction of experimental steel after hot rolling was experimentally examined by Vickers Indenter, TEST TECH, Shanghai, China.

## 3. Results

### 3.1. Ferrite/Pearlite Distribution and Partitioning of Solute Elements in the Cast Slab

Figure 2a shows the macrostructure of the 20CrMnTi billet. It can be seen that the heart of the 20CrMnTi gear steel circular billet is a thick uniform equiaxed crystal. Some columnar crystals appeared at 1/4 of the diameter, and a large number of tiny dendritic crystals were distributed at the edge. Figure 2b shows the dendrite morphology of the heart of the cast slab. Figure 2c shows the dendrite morphology of the 1/4 diameter of the cast slab. Figure 2d shows the dendrite morphology of the edge of the cast slab. According to the measurement, the equiaxed crystal size at the heart of the cast slab is 222 μm, the secondary dendrite spacing at the 1/4 diameter position of the cast slab is about 150 μm, and the secondary dendrite spacing at the edge position of the cast slab is 76 μm. The growth mode of solidified dendrites depends on the cooling intensity, the direction of heat dissipation, and the supercooling of the solidification front. When the components in the solidification front are too cold, the solidification front grows preferentially in the direction of dendritic quasi-heat dissipation, and is in the shape of columnar crystal. With the decrease of cooling intensity and the increase of subcooling of dendrite growth interface, equiaxial nucleation and growth occurred at the solidification front.

In order to further study the element segregation at different solidification positions of the cast slab, EPMA was used to perform surface scan analysis on the samples. Figure 2bi indicates the results for the sample at the heart of the cast slab, Figure 2ci indicates the results for the sample at the 1/4 diameter position of the cast slab, and Figure 2di indicates the results for the sample at the edge of the cast slab. Here, i = 1 for the SEM image, and i = 2, 3, 4, 5, and 6 for the elemental mappings of C, Cr, Mn, Si, and Ti, respectively. In Figure 2b1–b6, C concentrates into the pearlite phase; the dispersion of Cr, Mn and Si content is small except for topical concentration of Cr, Mn, and Si in the pearlite phase, while Ti is hardly partitioned between the ferrite and pearlite phase. In Figure 2c1–c6, C concentrates into the pearlite phase, the dispersion of Cr and Mn content is small except for topical concentration of Cr and Mn in the pearlite phase, while Si and Ti are hardly partitioned between the ferrite and pearlite phase. In Figure 2d1–d6, C, Cr, and Mn concentrate into the pearlite phase, while Si and Ti are hardly partitioned between the ferrite and pearlite phase. It can be seen from Figure 2bi–di that the segregation distribution of solute elements exists in different positions of the cast slab, and the degree of segregation distribution of solute elements is related to the dendrite morphology.

Figure 3 shows the dependence of structure and element segregation’s width on the cast slab’s position. The results for the dendrite ferrite phase and interdendritic pearlite phase are indicated in Figure 3a. The results for the segregation of C, Cr, Mn, and Si in the interdendritic pearlite phase are indicated in Figure 3b. As the position shifts from the heart to the edge position of the cast slab, the mean width of the dendrite ferrite phase and interdendritic pearlite phase decrease, and the mean width of C, Cr, Mn, and Si segregation in interdendritic pearlite phase decrease, too.

Figure 4 shows SEM images and partition of solute elements in ferrite and pearlite phase for the samples taken from different positions of the cast slab. Points 1–5 are the pearlite regions between dendrites, and points 6–10 are the ferrite regions of dendrites. Figure 4ai,bi,ci indicates the result for the samples at the heart, the 1/4 diameter position, and the edge of the cast slab, respectively. Furthermore, i = 1 for the SEM image, in which the numbers of 1–10 points are the scanning positions of the EPMA, and i = 2 for the partition of C, Si, Mn, and Cr in ferrite and pearlite phase. Here the mean content was obtained from a point analysis in a manner such that the content of each solute element was measured in five ferrite grains and five pearlite grains, respectively, and the error bar indicates the standard deviation. As can be seen, the content of C, Mn, and Cr in pearlite is higher than that in ferrite, and the content of si in ferrite is close to that in pearlite.

### 3.2. Ferrite-Bainite Banding Formation after Hot Rolling

The LSCM and SEM images of the samples after hot rolling are shown in Figure 5. Figure 5a–c indicates the LSCM images for the samples heated at 1100 °C for 2 h taken from the heart, the 1/4 diameter position, and the edge of the cast slab, respectively. Furthermore, i = 1 and 2 for the SEM image of local area organization. Figure 5a depicts an obvious banded structure after hot rolling of the sample taken from the heart of the cast slab. Figure 5a1 depicts the microstructure inside the red rectangle, and it can be found that it is bainite. Figure 5a2 depicts the microstructure inside the green rectangle, it is ferrite and bainite. The mean width of bainite banding in Figure 5a is measured to be 179 μm. Figure 5b depicts an obvious banded structure after hot rolling of the sample taken from the 1/4 diameter position of the cast slab. Figure 5b1 depicts the microstructure inside the red rectangle, and it can be found that it is bainite. Figure 5b2 depicts the microstructure inside the green rectangle, it is ferrite and bainite. The mean width of bainite banding in Figure 5b is measured to be 65 μm, and it is narrower than that in Figure 5a,c, which depicts structure after hot rolling of the sample taken from edge of the cast slab, and there is no banded structure. Figure 5c1 depicts the microstructure inside the red rectangle, it is bainite. Figure 5b2 depicts the microstructure inside the green rectangle, it is Ferrite. Figure 5c shows the uniform distribution of ferrite and bainite structure for the sample taken from the edge of the cast slab after hot rolling.

Figure 5d–f indicates the LSCM images for the samples heated at 1150 °C for 2 h, 1200 °C for 2 h, and 1200 °C for 4 h, respectively, and they were all taken from the 1/4 diameter position of the cast slab. Figure 5d depicts a slight banded structure of the sample heated at 1150 °C for 2 h. Figure 5d1 depicts the microstructure inside the red rectangle, and it can be found that it is bainite. Figure 5d2 depicts the microstructure inside the green rectangle, it is ferrite and bainite. The width of bainite banding in Figure 5d is measured to be 51 μm, and it is narrower than that in Figure 5b. Figure 5e depicts uniform distribution structure for the sample heated at 1200 °C for 2 h, without a banded structure. Figure 5e1,e2 is an enlarged view of the organization of Figure 5e, it shows uniform distribution of the bainite structure. Figure 5f depicts a uniform distribution structure for the sample heated at 1200 °C for 4 h, without a banded structure, too. Figure 5f1,f2 is an enlarged view of the organization of Figure 5f, it shows uniform distribution of the bainite structure, and the bainite grain size in the sample heated at 1200 °C for 4 h is larger than that in the sample heated at 1200 °C for 2 h.

Figure 6 shows SEM images and elemental mappings of the solute elements for the samples heated at 1100 °C for 2 h during hot rolling. Figure 6 ai, bi indicates the results for samples taken from the heart and 1/4 diameter position of the cast slab, respectively. Furthermore, i = 1 for the SEM image, and i = 2, 3, 4, 5, and 6 for the elemental mappings of C, Cr, Mn, Si, and Ti, respectively. In Figure 6a1–a6, the distribution of C and Mn in the bainite phase are rather homogeneous, but that of Si and Cr in the bainite phase is slightly inhomogeneous, while Ti is hardly partitioned between the ferrite and bainite phase, without segregation distribution. After measurement, it is found that the solute element banding segregation widths of C, Cr, Mn, and Si are 182 μm, 104 μm, 160 μm, and 157 μm, respectively. In Figure 6b1–b6, the distribution of C and Mn in the bainite phase are rather homogeneous, but that of Cr and Si in the bainite phase is slightly inhomogeneous, while Ti is hardly partitioned between the ferrite and bainite phase, without segregation distribution. The solute element banding segregation widths of C, Cr, Mn, and Si are 44 μm, 52 μm, 60 μm, and 70 μm. The element banding segregation width of the sample taken from the 1/4 diameter position is lower than that of the sample taken from the heart of the cast slab after same hot rolling process.

Figure 7 shows the hardness distribution along the normal direction of samples after hot rolling. Figure 7a–c indicates the hardness of samples heated at 1100 °C for 2 h during the hot rolling process taken from the heart, 1/4 diameter position, and edge of the cast slab, respectively. Figure 7a indicates the unevenly distributed hardness of the sample taken from the heart of the cast slab along the normal direction, and the average of it is 317 HV. Figure 7b indicates the hardness along the normal direction of the sample taken from the 1/4 diameter position of the cast slab, which fluctuates slightly, and the average of it is 271 HV. Figure 7c indicates the evenly distributed hardness along the normal direction of the sample taken from the edge of the cast slab, and the average of it is 217 HV. After the same hot rolling process, the hardness of the sample taken from the heart of the cast slab is the largest, and the distribution of it is the most uneven. The banded structure is the most serious and the content of bainite is the highest. The hardness of the sample taken from the 1/4 diameter position of the cast slab decreases, and the uneven distribution of it decreases. The banded structure is less and the content of bainite is reduced. The hardness of the sample taken from the edge of the cast slab is the lowest, and the distribution of it is the most uniform. Because it has no banded structure, and the ferrite and bainite organization are evenly distributed, the content of bainite organization is the lowest.

Figure 7d–f indicates the hardness of samples taken from the 1/4 diameter position of the cast slab heated at 1150 °C for 2 h, 1200 °C for 2 h, and 1200 °C for 4 h during hot rolling process, respectively. Figure 7d indicates the unevenly distributed hardness of the sample heated at 1150 °C for 2 h along the normal direction, and the average of it is 320 HV. Figure 7e indicates the hardness along the normal direction of the sample heated at 1200 ℃ for 4 h, which distributes evenly, and the average of it is 317 HV. Figure 7f indicates the evenly distributed hardness along the normal direction of the sample heated at 1200 °C for 4 h, and the average of it is 312 HV. The average hardness of the three samples under different hot rolling processes are very close, because all of them have a lot of bainite organization. With the increase of heating temperature and the extension of heating time, the hardness distribution becomes more and more uniform, because the organization distributes more and more evenly.

## 4. Discussion

### 4.1. The Influence of Element Segregation and Diffusion on the γ→α Transformation Temperature

In the continuous casting process, the liquid steel grows in the form of dendrites during solidification. K_0_ is the equilibrium distribution coefficient of the solute between liquid and solid. Components with K_0_ < 1 are always easy to be enriched in grain boundaries or between dendrites, so the concentration of this element in the interdendritic region is significantly higher than that in the dendritic region concentration.

In order to better explain the microstructure evolution during the hot rolling process in the presence of element segregation, diffusion of solute elements during heating was simulated by DICTRA with TCFE10 and MOBFE5 databases. The size of the simulation cell was set to 111 μm, 75 μm, and 38 μm, respectively, as half of the mean distance between the centers of adjacent ferrite and pearlite for samples taken from the heart, 1/4 diameter position, and edge of the cast slab. As shown in Figure 4, according to the measurement by EPMA, C, Cr, Mn, and Si were partitioned between the ferrite and pearlite phases in the cell. For the simulation of T = 1100 °C, 1150 °C, and 1200 °C, the initial microstructure was set as γ one-phase microstructure. The profiles of the C, Cr, Mn, and Si contents in the cell are shown in Figure 8. Figure 8ai,bi,ci indicates solute elements profiles of the sample heated at 1100 °C taken from the heart, 1/4 diameter position, and edge of the cast slab, respectively, and i = 1, 2, 3, and 4 refer to element contents of C, Cr, Mn, and Si, respectively. It can be seen from the simulation result of Figure 8a1,b1,c1 that after heating at 1100 °C for 2 h, the element C diffused greatly and tended to be evenly distributed. Figure 8a2,b2,c2 indicates that the element Cr diffuses slightly after heating at 1100 °C for 2 h, but the Cr distributed more evenly in the sample taken from the edge of the cast slab than samples taken from the 1/4 diameter position and heart of the cast slab. It indicates that the more uniform the initial segregation, the easier for element Cr to diffuse uniformly. Figure 8a3,b3,c3 indicates that the degree of segregation of element Mn becomes greater. Figure 8a4,b4,c4 indicates that the element Si diffuses a lot more than elements Cr and Mn, and the more uniform the initial segregation, the easier for element Si to diffuse uniformly.

Figure 8bi,di,ei indicates solute elements profiles of samples taken from the 1/4 diameter position of the cast slab heated at 1100 °C, 1150 °C, and 1200 °C, respectively, and i = 1, 2, 3, and 4 refer to element contents of C, Cr, Mn, and Si, respectively. It can be seen from the simulation result of Figure 8b1,d1,e1 that after heating the element C diffused greatly and tended to be evenly distributed. Figure 8b2,d2,e2 indicates that as the heating temperature increases from 1100 °C to 1200 °C, the element Cr diffuses more uniformly. Figure 8e indicates that as the heating time increases from 2 h to 4 h, the element Cr diffuses more uniformly. Figure 8b3,d3,e3 indicates that the segregation of element Mn of the sample heated at 1100–1200 °C for 2 h becomes greater, however, it becomes smaller after being heated at 1200 °C for 4 h. Figure 8b4,d4,e4 indicates that the element Si diffuses a lot more than elements Cr and Mn, and the higher the temperature, the longer the heating time, and the easier for element Si to diffuse uniformly.

The segregation coefficient (K) indicates the degree of segregation of solute elements, and K is the ratio of mass fractions of interdendritic elements to dendritic elements (take the average). The closer K is to 1, the smaller the segregation and the higher the degree of homogenization. When the diffusion is uniform, the value of K becomes 1. The K of solute elements for samples taken from heart, 1/4 diameter position, and edge of the cast slab are shown in Figure 9a. Figure 9ai indicates the K of solute elements for initial samples taken from the cast slab and samples heated at 1100 °C for 2 h, and C, Cr, Mn, and Si for i = 1, 2, 3, and 4, respectively. Figure 9a1 indicates that the K of C for the sample taken from the heart of the cast slab is maximum, and the K of C for the sample taken from the 1/4 diameter position of the cast slab is smaller, and the K of C for the sample taken from the edge of the cast slab is minimum. What is more, after being heated at 1100 °C for 2 h, the K of C for all those three samples taken from different positions of the cast slab becomes 1. Figure 9a2 indicates that the K of Cr for the sample taken from the heart of the cast slab is maximum, for the sample taken from the 1/4 diameter position of the cast slab is smaller than it, and for the sample taken from the edge of the cast slab is minimum. What is more, after being heated at 1100 °C for 2 h, the K of Cr for all those three samples taken from different positions of the cast slab all become smaller and kept the same initial rules. Figure 9a3 indicates the K of Mn for the sample taken from the heart of the cast slab is maximum, for the sample taken from the 1/4 diameter position of the cast slab is smaller than it, and for the sample taken from the edge of the cast slab is minimum. However, after being heated at 1100 °C for 2 h, the K of Mn for the sample is taken at the heart of the cast slab constant as initial, the sample taken from the 1/4 diameter position of the cast slab gets bigger, and the sample taken from the edge of the cast slab gets a little smaller. Figure 9a4 indicates that the K of Si for samples taken from different positions of the cast slab are close to 1. What is more, after being heated at 1100 °C for 2 h, the K of Si for all those three samples taken from different positions of the cast slab all become closer to 1. In summary, after being heated at 1100 °C for 2 h, the C element in the samples at different positions diffused greatly and tended to be evenly distributed, and the segregation coefficient K was close to 1. However, the Si, Cr, and Mn element only diffuse slightly, and the segregation coefficient changes very little. Even the Mn element in the sample taken from the 1/4 diameter position of the cast slab was still in the upslope diffusion stage.

The K of solute elements for samples taken from the 1/4 diameter position of the cast slab heated at 1100–1200 °C for 2–4 h are shown in Figure 9b. Figure 9bi indicates the K of solute elements for initial samples and samples after heating, and C, Cr, Mn, and Si for i = 1, 2, 3, and 4, respectively. Figure 9b1 indicates that the K of C for the initial sample is maximum, and for the sample heated at 1100–1200 °C for 2–4 h, all become 1. Figure 9b2 indicates that the K of Cr for the initial sample is maximum, and for the sample heated at 1100–1200 °C for 2–4 h, all become smaller; what is more, with the increase of heating temperature and extension of heating time, K becomes smaller and smaller. Figure 9b3 indicates that the K of Mn for the sample heated at 1100–1200 °C for 2 h, all become larger than the initial, and the sample heated at 1200 °C for 4 h becomes smaller than the initial. While with the increase of heating temperature and extension of heating time, K becomes smaller and smaller. Figure 9b4 indicates that the K of Si for the initial sample is minimum and less than 1, and for the sample heated at 1100–1200 °C for 2–4 h, all become larger and closer to 1. What is more, with the increase of heating temperature and extension of heating time, K becomes larger and closer to 1.

The diffusion coefficient of carbon in solid steel is much higher than that of other elements [32]. In austenite, carbon can be approximately distributed in thermodynamic equilibrium, while the diffusion coefficients of alloying elements such as Si, Cr, and Mn are less than that of carbon, and dendrite segregation is difficult to be eliminated by element diffusion. As shown by the simulation results in Figure 9, the diffusion of different solute elements varies greatly for samples with various initial segregations or after different heating processes. C has a large degree of diffusion and tends to be evenly distributed, but Cr, Mn, and Si have a small degree of diffusion, and Mn is still in the uphill diffusion stage. Regarding the temperature of sample drop from the austenitizing temperature to transformation temperature A_e3_ (γ→α transition temperature) along the cooling process, the segregation of solute elements will result in uneven distribution of C in austenite. Mn, Cr can effectively attract C, but Si is exclusive to C, so it will significantly change local austenitic, ferritic phase transformation A_e3_ temperature. In addition, the solute elements themselves will also affect the temperature of A_e3_.

With the help of Matlab calculation software, the equilibrium starting temperature of the phase transition is calculated through the super-component model and the thermodynamic model invented by Kaufman, Radcliffe, and Cohen (KRC model). The expression of the free energy of phase transition of the super-component model is shown in Equation (1). The calculation method of phase transition equilibrium temperature is: Let $\Delta G_{S}^{\gamma \to \alpha +\gamma_{1}}$ = 0, and the obtained temperature is A_e3_. The flow chart of the calculation model for γ→α phase transition equilibrium starting temperature is shown in Figure 10.

The γ→α phase transition temperature A_e3_ in the solute-rich zone and the solute-poor zone of the sample alloy elements can be calculated separately, and the γ→α phase transition temperature difference ΔA_e3_ between the solute-rich zone and the solute-poor zone can be calculated. Figure 11a shows A_e3_ and ΔA_e3_ of samples taken from heart, 1/4 diameter position, and edge of the cast slab, after being heated at 1100 °C for 2 h. From the calculation results, it can be seen that A_e3_ of the solute-poor region are higher than that of the solute-rich region for samples taken from the heart, 1/4 diameter position, and edge of the cast slab. ΔA_e3_ of the sample taken from heart of the cast slab is 18 °C, ΔA_e3_ of the sample taken from 1/4 diameter position of the cast slab is 17 °C, and ΔA_e3_ of the sample taken from the edge of the cast slab is 14 °C. It can be seen from Figure 5a–c, after being heated at 110 °C for 2 h during hot rolling, an obvious banded structure appeared in the samples taken from the heart and 1/4 diameter position of the cast slab, but there is no banded structure in the sample taken from edge of the cast slab.

Figure 11b shows A_e3_ and ΔA_e3_ of samples taken from the 1/4 diameter position after being heated at 1100–1200 °C for 2–4h. From Figure 11b, it can be seen that A_e3_ of the solute-poor region is higher than that of the solute-rich region in the samples. ΔA_e3_ of the sample heated at 1100 °C for 2 h is 17 °C, ΔA_e3_ of the sample heated at 1150 °C for 2 h is 15 °C, ΔA_e3_ of the sample heated at 1200 °C for 2 h is 13 °C, and ΔA_e3_ of the sample heated at 1200 °C for 4 h is 11 °C. It can be seen from Figure 5b,d–f, taken from 1/4 diameter of the cast slab, that an obvious banded structure appeared in the samples heated at 1100–1150 °C for 2 h, but there is no banded structure in the sample heated at 1200 °C for 2–4 h.
(1)$${\Delta G}_{s}^{\gamma \to \alpha }=141\sum x_{i}^{\alpha /\gamma }\left({\Delta T}_{M}^{i}-{\Delta T}_{\mathrm{NM}}^{i}\right)+{\Delta G}_{\mathrm{Fe}}^{\gamma \to \alpha }\left\{T-100\sum x_{i}^{\alpha /\gamma }{\Delta T}_{M}^{i}\right\}$$

In the formula, ${\Delta G}_{\mathrm{Fe}}^{\gamma \to \alpha }$ is the free energy change of γ→α phase transition of pure Fe;

The temperature shift $\Delta T_{M}^{i}$ is used to express the contribution of the substitutional alloying element X_i_ to the free energy difference ΔG of the ferromagnetic part during phase transformation;

The temperature shift $\Delta T_{NM}^{i}$ is used to express the contribution of the substitutional alloying element X_i_ to the free energy difference ΔG of the non-ferromagnetic part during phase transformation;

$x_{i}^{\alpha /\gamma }$ is the molar fraction of the replacement alloying element X_i_ on the ferrite side at the ferrite/austenite phase interface.

Therefore, it can be concluded that when the ΔA_e3_ is above 14 °C, the sample will appear to have a banded structure after hot rolling. When the ΔA_e3_ is below 14 °C, the sample will not appear to have a banded structure after hot rolling. The phase transition temperature of different regions is different. Proeutectoid ferrite preferentially precipitates in the position with low carbon content and high A_e3_. When the cooling rate is constant, the greater the ΔA_e3_, the greater the difference in the phase transition time between the low-concentration zone and the high-concentration zone, so that the C has sufficient diffusion time from the proeutectoid ferrite zone to the adjacent zone. The segregation zone in the austenite state further causes uneven distribution of carbon. As shown in Figure 2, Figure 3, Figure 4, Figure 5 and Figure 6, the more serious the dendrite segregation, the more uneven the solute element distribution and the more serious the banded structure. The higher the heating temperature and the longer the heating time, the more uniform the element diffusion and the less obvious the band structure. As shown in Figure 7, corresponding to the banding structure, the more severe the dendrite segregation, the more uneven the hardness distribution. The higher the heating temperature and the longer the heating time, the more uniform the hardness distribution.

### 4.2. The Role of Austenite Grain Size in the Ferrite-Bainite Banding Formation during Hot Rolling

Figure 12 indicates the LSCM of austenite grain morphology for samples cooled in water after hot rolling, and they are all taken from 1/4 diameter of the cast slab. Figure 12a indicates austenite grain morphology for samples heated at 1100 °C for 2 h, and the austenite grain is fine, which is 18 μm after measurement. Figure 12b indicates austenite grain morphology for samples heated at 1150 °C for 2 h, and the austenite grain is fine, which is 21 μm after measurement. Figure 12c indicates austenite grain morphology for samples heated at 1200 °C for 2 h, and the austenite grain is obviously coarser than that of the samples heated at 1100–1200 °C for 2 h, which is 27 μm after measurement. Figure 12d indicates austenite grain morphology for samples heated at 1200 °C for 4 h, and the austenite grain is coarser, which is 33 μm after measurement.

Figure 13 indicates a relationship between austenite grain and heating temperature and heating time for samples taken from 1/4 diameter of the cast slab. As the heating temperature increases, the austenite grain size becomes larger. With the extension of heating time, the austenite grain size becomes larger. It can be seen from Figure 5b, d–f, taken from 1/4 diameter of the cast slab, an obvious banded structure appeared in the samples heated at 1100–1150 °C for 2 h, but there is no banded structure in samples heated at 1200 °C for 2–4 h. Therefore, it can be concluded that when the austenite grain size is above 21 μm, the sample will not appear to have a banded structure after hot rolling. When the austenite grain size is below 21 μm, the sample will appear to have a banded structure after hot rolling.

As ferrite nucleates at austenite grain boundaries in low Mn regions, carbon is then rejected into the remaining austenite. Thus, the travel distance for carbon that results in severe banding is directly related to austenite grain size. Thompson and Howell proposed the importance of austenite grain size to the formation of banded structures in 1992. They proposed that when the austenite grains are small, the ferrite grains are in the austenite region with high A_e3_ transformation temperature. Nucleation is formed at the bulk grain boundary, and these grains collide with each other to grow together in the rolling direction and finally connect together to form a plane extending to the area with high Mn content. However, if the austenite grains are significantly larger than the spacing of the chemical segregation zones, the ferrite-pearlite banded structure is not easy to produce. When there are insufficient (Mn, Fe) S and other secondary particles, ferrite can only preferentially nucleate at the austenite grain boundary crossing the low-manganese zone. In order to form a ferrite banding, the nucleation positions of individual ferrites must be relatively close to ensure that they can be connected to each other after growth. However, the austenite grains are large, and there is no sufficient nucleation site for ferrite, and it is difficult for many single ferrite structures to connect to each other into a plane, and it is impossible to form a ferrite banding. Experiments show that when the austenite grain size is greater than 2 to 3× the spacing of the chemical segregation zone, the band structure will not be formed.

## 5. Conclusions

The partitioning and diffusion of solute elements in hot rolling and the effect of the partitioning and diffusion on the ferrite-bainite banding formation after hot rolling in the 20CrMnTi steel were experimentally examined by EPMA technology and simulated by DICTRTA and MATLAB software. The austenite grain size related to the hot rolling process and the effect of austenite grain size on the ferrite-bainite banding formation were studied. The hardness along the normal direction of samples after hot rolling was experimentally examined by Vickers Indenter. The main conclusions are summarized as follows.

(1)Samples without banding have the most uniform hardness distribution, which are taken from edge of the cast slab and 1/4 diameter position of the cast slab, heating at 1100 °C for 2 h and above 1200 °C for 2–4 h during the hot rolling, respectively.(2)In the cast slab, solute elements are segregated and distribute in ferrite and pearlite. The segregation width and segregation coefficient K of elements for samples taken from the heart of the cast slab are larger than those that taken from the 1/4 diameter position and edge of the cast slab. Cr, Mn, and Si diffuse and inhomogeneously concentrate in austenite during hot rolling, while C homogeneously concentrates in austenite. After the same hot rolling process, ΔA_e3_ increases and ferrite-bainite banding intensifies with increasing initial segregation width and segregation coefficient K of solute elements. Under the same initial segregation of solute elements, ΔA_e3_ drops and ferrite-bainite banding reduces with increasing heating temperature and extension heating time. To sum up, when ΔA_e3_ drops below 14 °C, ferrite-bainite banding even disappears.(3)The austenite grain size increase with increasing heating temperature and extension heating time. It can be concluded that when the austenite grain size is above 21 μm, the sample will not appear to have a banded structure after hot rolling.

## Figures and Tables

**Figure 1 materials-14-02356-f001:**
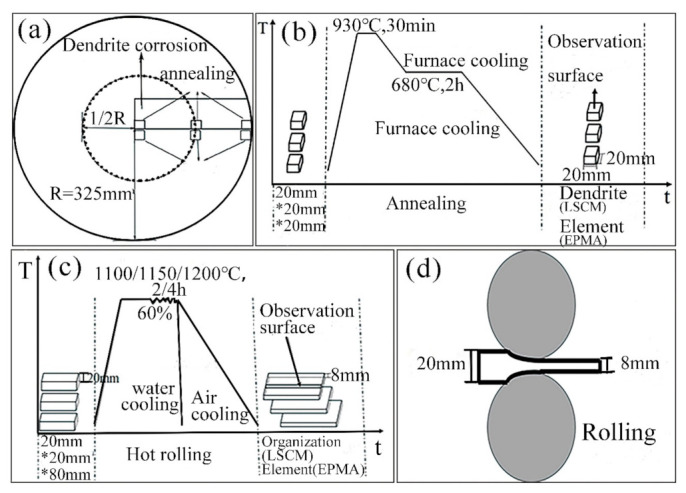
(**a**) Sampling position, (**b**) annealing process of casting slab erosion dendrite sample, (**c**) hot rolling process, (**d**) rolling schematic diagram.

**Figure 2 materials-14-02356-f002:**
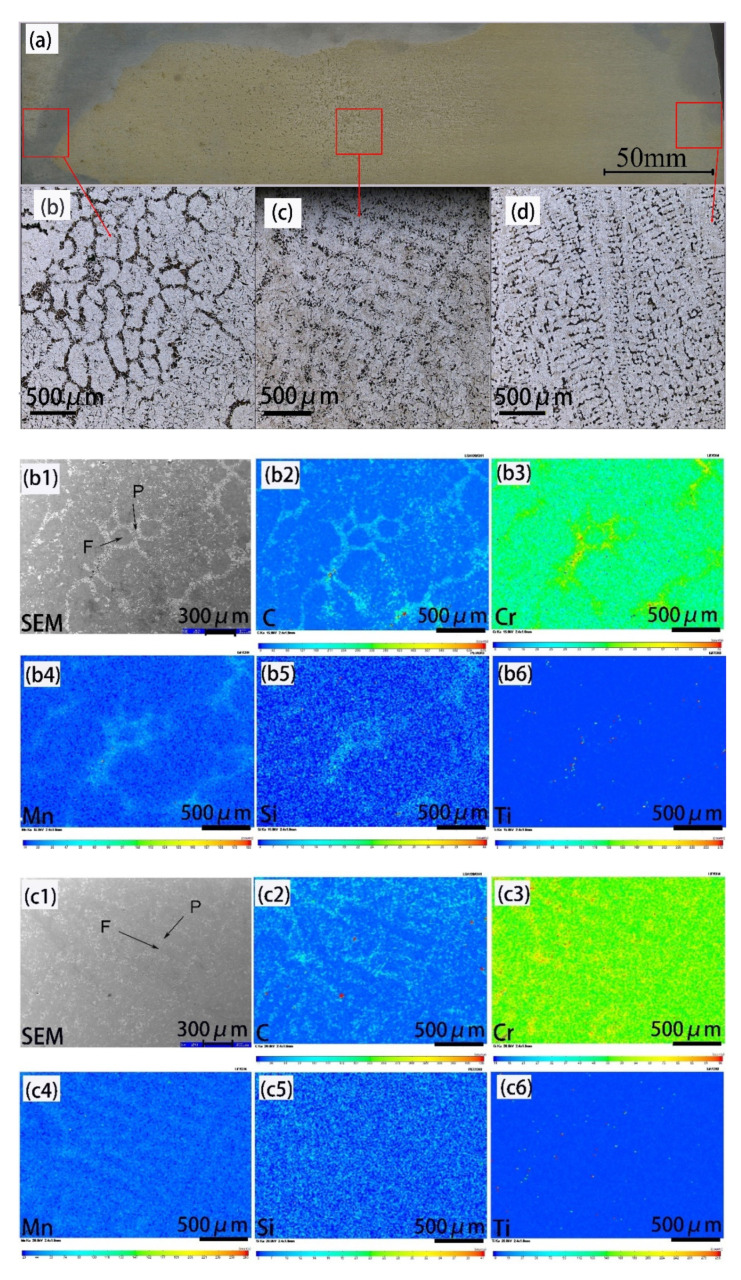
Cast slab solidification structure and the distribution of solute elements. (**a**) The distribution of dendrites in the radius direction; (**b**) the dendrite morphology of the sample at the heart of the cast slab; and (**c**) the dendrite morphology of the sample at the 1/4 diameter position of the cast slab; (**d**) dendrite morphology of sample at the edge of the cast slab; (**b1**–**b6**) SEM images and elemental mappings of C, Cr, Mn, Si, and Ti for sample taken from the heart of the cast slab, respectively; (**c1**–**c6**) SEM images and elemental mappings of C, Cr, Mn, Si, and Ti for sample taken from the 1/4 diameter position of the cast slab, respectively; (**d1**–**d6**) SEM images and elemental mappings of C, Cr, Mn, Si, and Ti for sample taken from the edge of the cast slab, respectively.

**Figure 3 materials-14-02356-f003:**
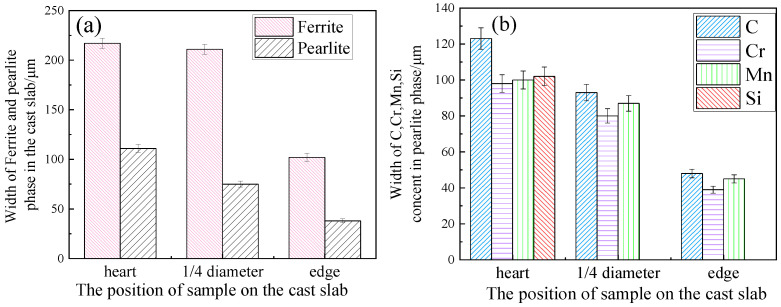
Structure and element segregation’s width of samples. (**a**) The widths of dendrite ferrite phase and interdendritic pearlite phase, with error bars. (**b**) The width of segregation zone for C, Cr, Mn, and Si in interdendritic pearlite phase, with error bars.

**Figure 4 materials-14-02356-f004:**
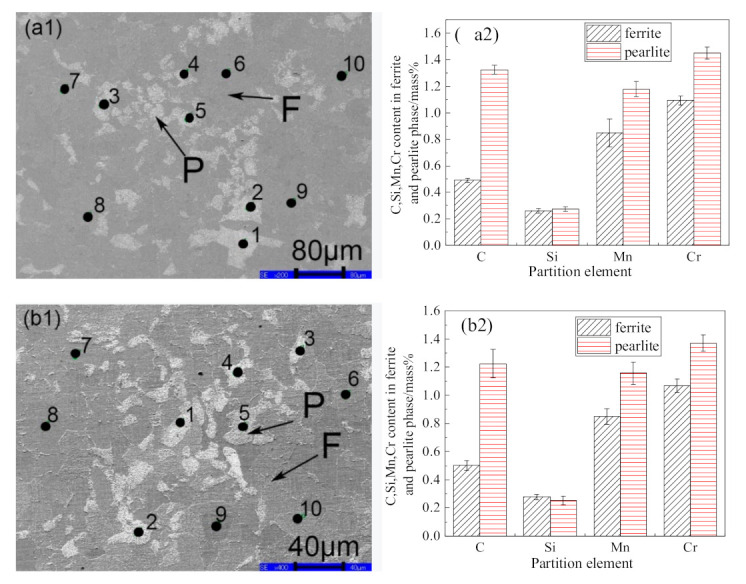
(**a1**–**c1**) SEM images and (**a2**–**c2**) partition of C, Si, Mn, and Cr between ferrite and pearlite phase for samples at the (**a**) heart, (**b**) 1/4 diameter position, and (**c**) edge of the cast slab.

**Figure 5 materials-14-02356-f005:**
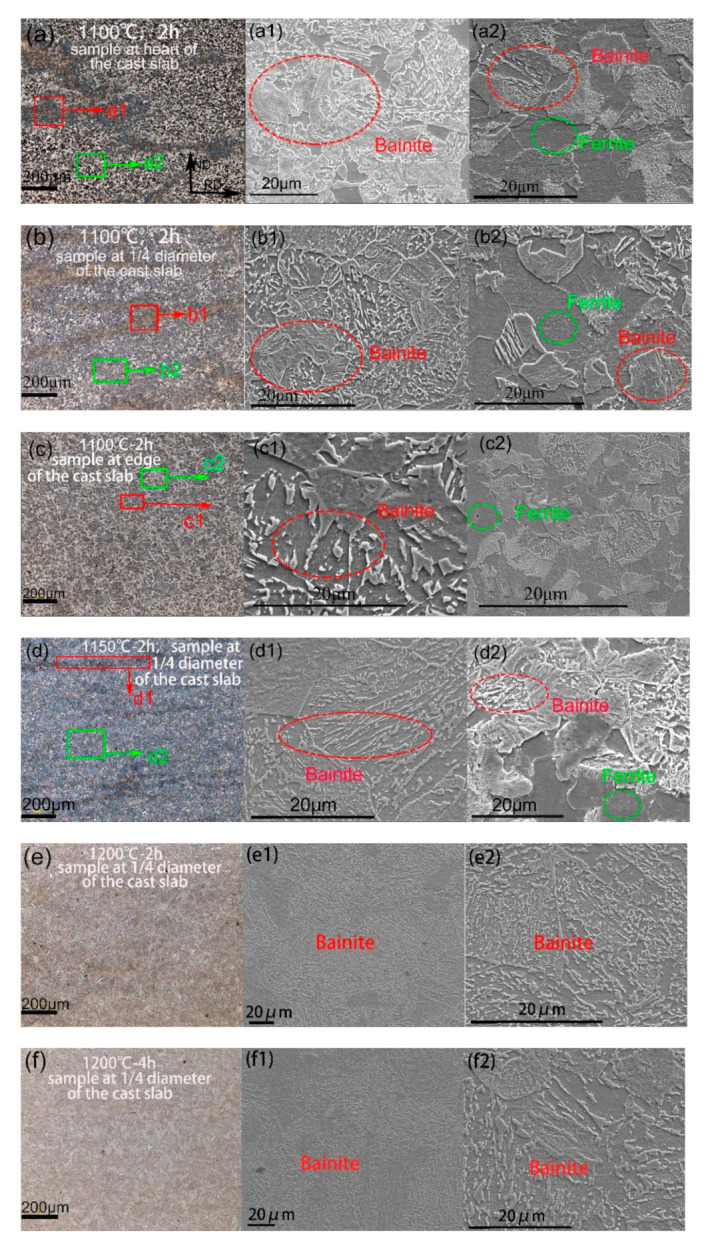
LSCM of samples heated at 1100 °C for 2 h during the hot rolling process taken from (**a**) heart, (**b**) 1/4 diameter position, (**c**) the edge of the cast slab, respectively, and samples taken from the 1/4 diameter position of the cast slab heated at (**d**) 1150 °C for 2 h, (**e**) 1200 °C for 2 h, and (**f**) 1200 °C for 4 h during the hot rolling process, respectively. SEM images of samples heated at 1100 °C for 2 h during the hot rolling process taken from (**a1**,**a2**) heart, (**b1**,**b2**) 1/4 diameter position, (**c1**,**c2**) the edge of the cast slab, respectively, and samples taken from the 1/4 diameter position of the cast slab heated at (**d1**,**d2**) 1150 °C for 2 h, (**e1**,**e2**) 1200 °C for 2 h, and (**f1**,**f2**) 1200 °C for 4 h during the hot rolling process, respectively.

**Figure 6 materials-14-02356-f006:**
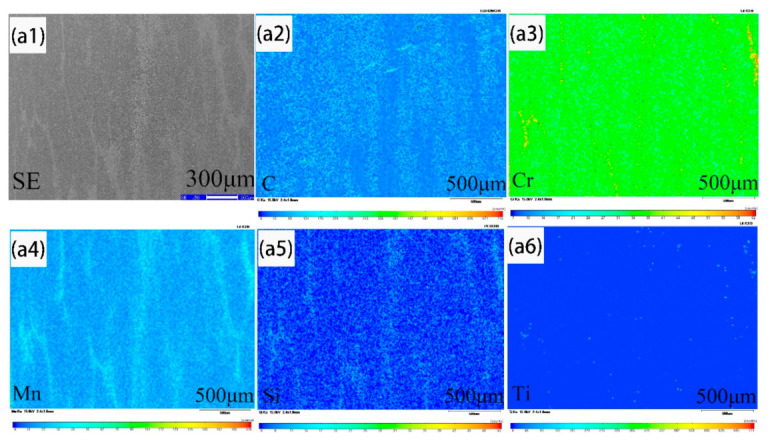
Solute element distribution of the samples heated at 1100 °C for 2 h after rolling. (**a1**–**a6**) SEM images and elemental mappings of C, Cr, Mn, Si, and Ti for sample taken from the heart of the cast slab; (**b1**–**b6**) SEM images and elemental mappings of C, Cr, Mn, Si, and Ti for sample taken from the 1/4 diameter of the cast slab, respectively.

**Figure 7 materials-14-02356-f007:**
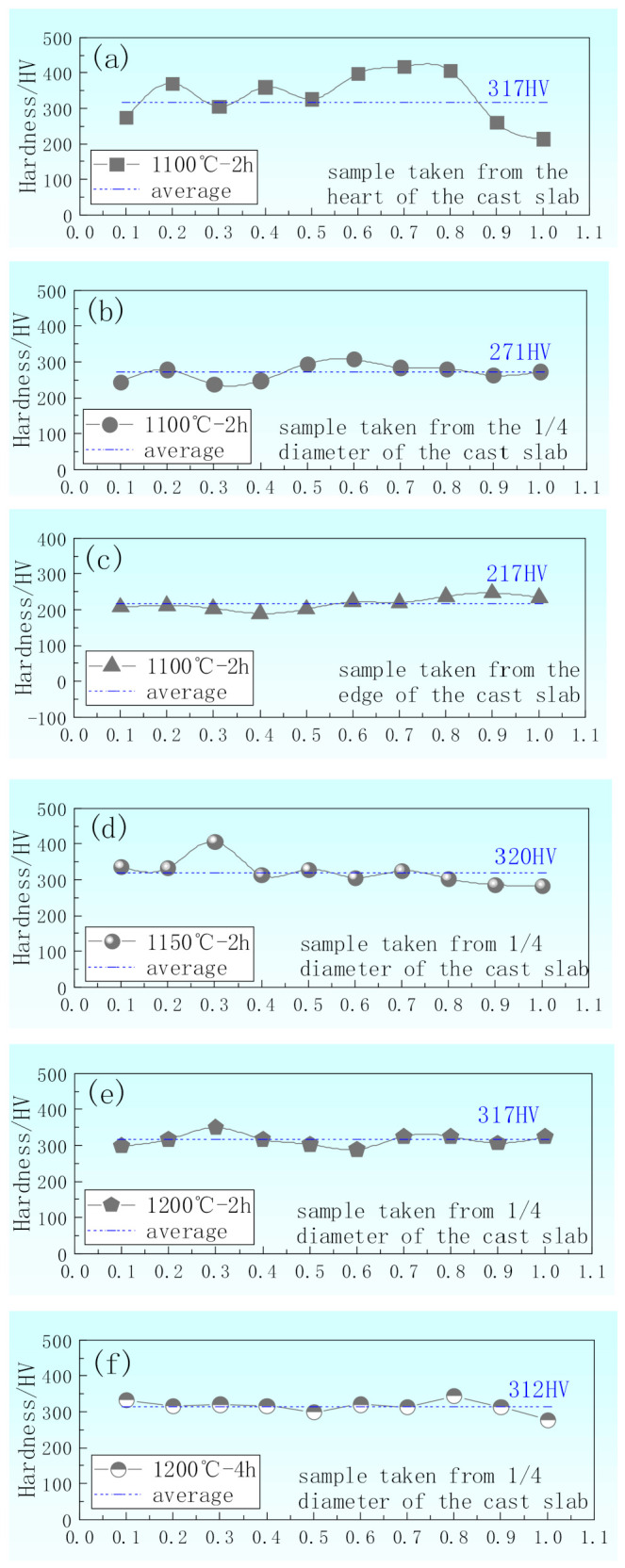
Hardness distribution along the normal direction of samples after hot rolling. Hardness of samples heated at 1100 °C for 2 h during the hot rolling process taken from the (**a**) heart, (**b**) 1/4 diameter position, and (**c**) edge of the cast slab, respectively, and samples taken from the 1/4 diameter of the cast slab heated at (**d**) 1150 °C for 2 h, (**e**) 1200 °C for 2 h, and (**f**) 1200 °C for 4 h during the hot rolling process, respectively.

**Figure 8 materials-14-02356-f008:**
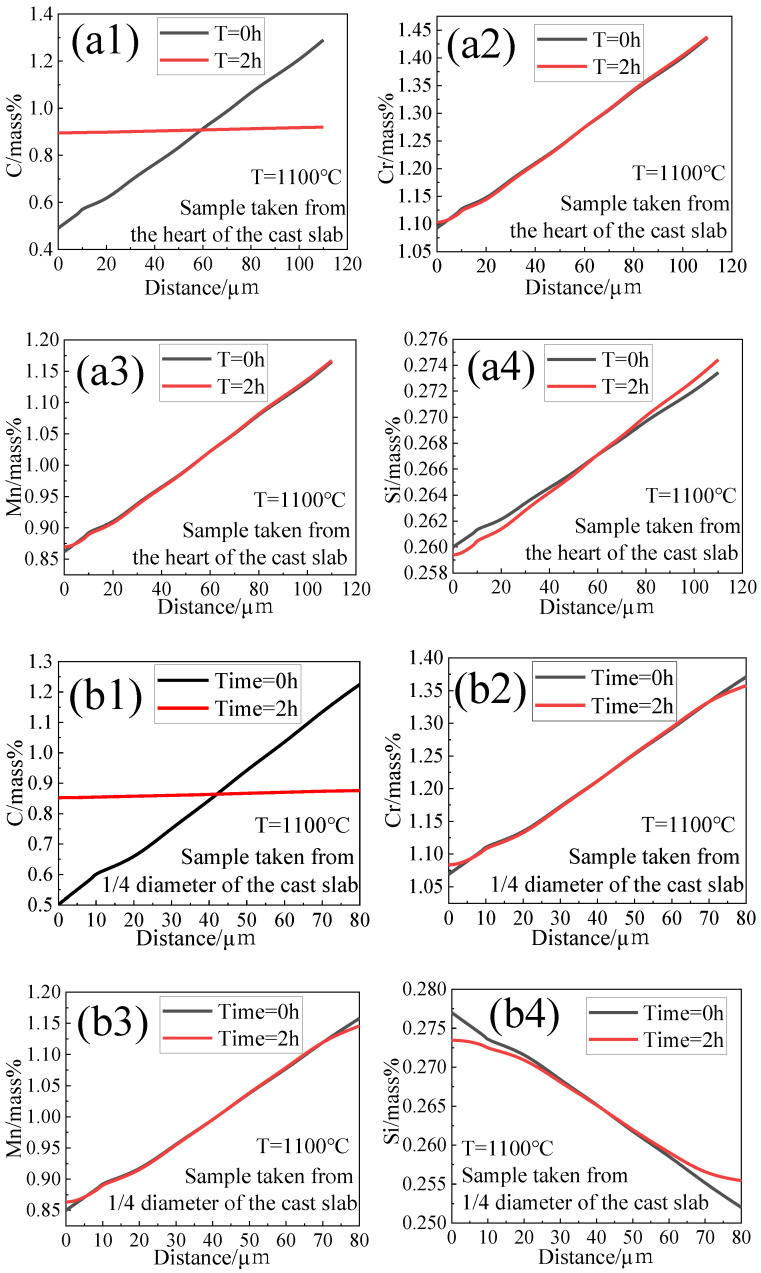
Profiles of C, Cr, Mn, and Si contents at various initial segregations and various heating process calculations by DICTRA. (**a1**–**a4**) C, Cr, Mn, and Si contents of samples taken from the heart of the cast slab heated at 1100 °C for 2 h, respectively; (**b1**–**b4**) C, Cr, Mn, and Si contents of samples taken from the 1/4 diameter of the cast slab heated at 1100 °C for 2 h, respectively; (**c1**–**c4**) C, Cr, Mn, and Si contents of samples taken from the edge of the cast slab heated at 1100 °C for 2 h, respectively; (**d1**–**d4**) C, Cr, Mn, and Si contents of samples taken from the 1/4 diameter of the cast slab heated at 1150 °C for 2 h, respectively; (**e1**–**e4**) C, Cr, Mn, and Si contents of samples taken from the 1/4 diameter of the cast slab heated at 1200 °C for 2~4h, respectively.

**Figure 9 materials-14-02356-f009:**
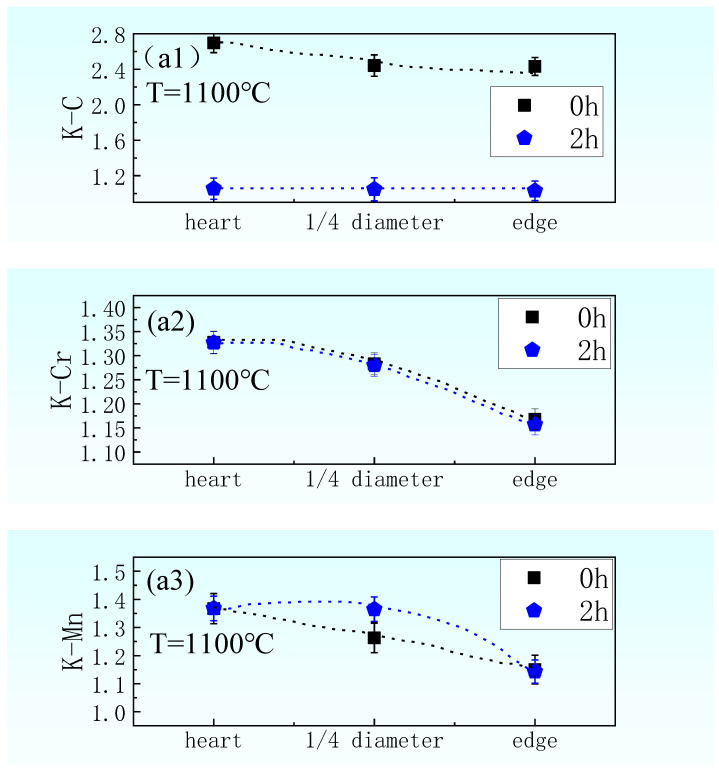
The segregation coefficient K of elements C, Cr, Mn, and Si at various initial segregations and after various heating processes. (**a1**–**a4**) The segregation coefficients K of C, Cr, Mn, and Si for the initial samples taken from the cast slab and samples heated at 1100 °C for 2 h, with error bars; (**b1**–**b4**) The segregation coefficients K of C, Cr, Mn, and Si for samples taken from the 1/4 diameter position of the cast slab heated at 1100–1200 °C for 2–4 h, with error bars.

**Figure 10 materials-14-02356-f010:**
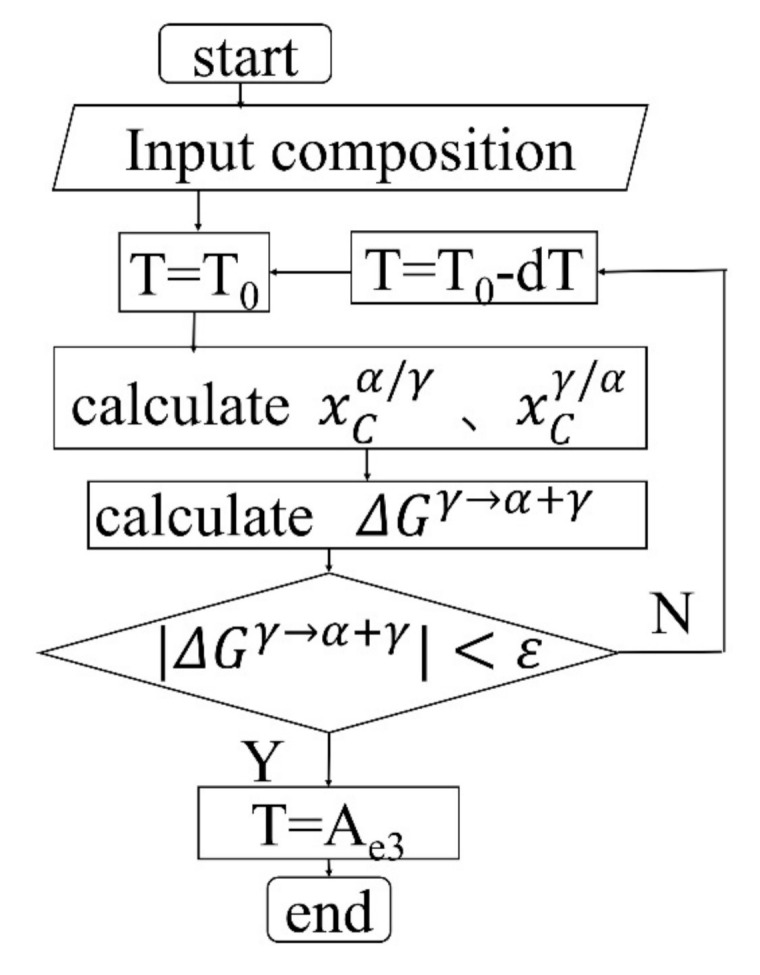
γ→α phase transition equilibrium starting temperature calculation model.

**Figure 11 materials-14-02356-f011:**
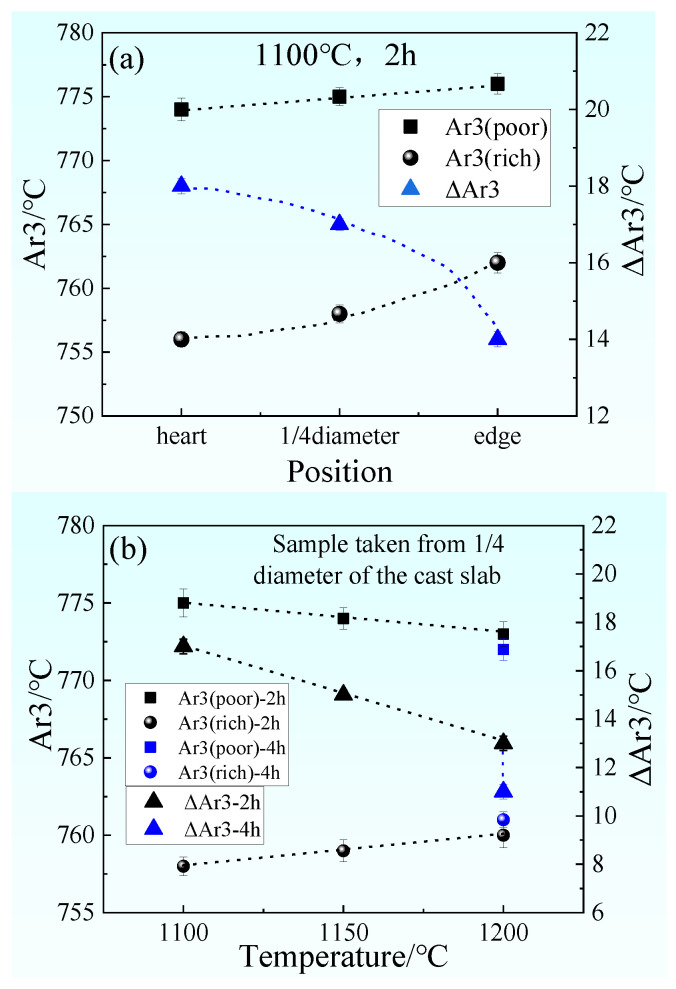
The γ→α phase transition temperature A_e3_ and the difference of γ→α phase transition temperature ΔA_e3_ of the solute-rich zone and solute-poor zone for samples at various initial segregations and after various heating process. (**a**) A_e3_ and ΔA_e3_ for samples taken from different positions of the cast slab, after being heated at 1100 °C for 2 h. (**b**) A_e3_ and ΔA_e3_ for samples heated at different temperatures for different time, taken from the 1/4 diameter position of the cast slab, with error bars.

**Figure 12 materials-14-02356-f012:**
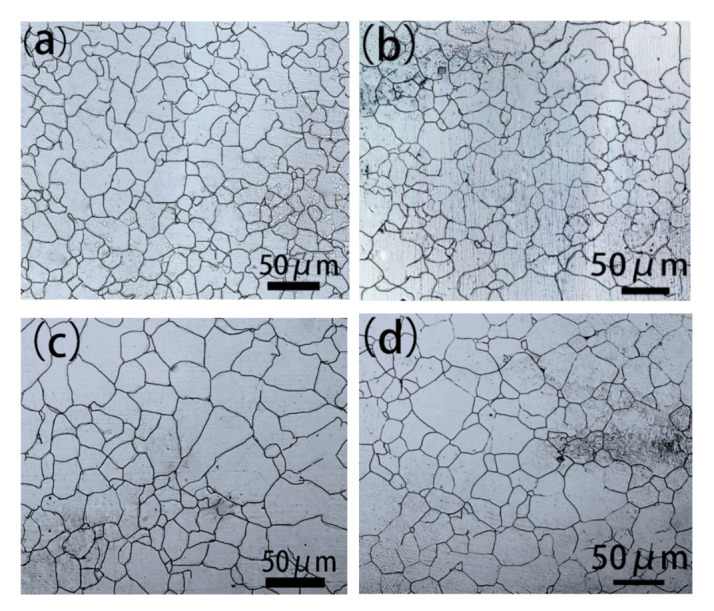
LSCM of austenite grain for samples cooled in water, which are heated at (**a**) 1100 °C for 2, (**b**) 1150 °C for 2 h, (**c**) 1200 °C for 2 h, and (**d**) 1200 °C for 4 h, respectively.

**Figure 13 materials-14-02356-f013:**
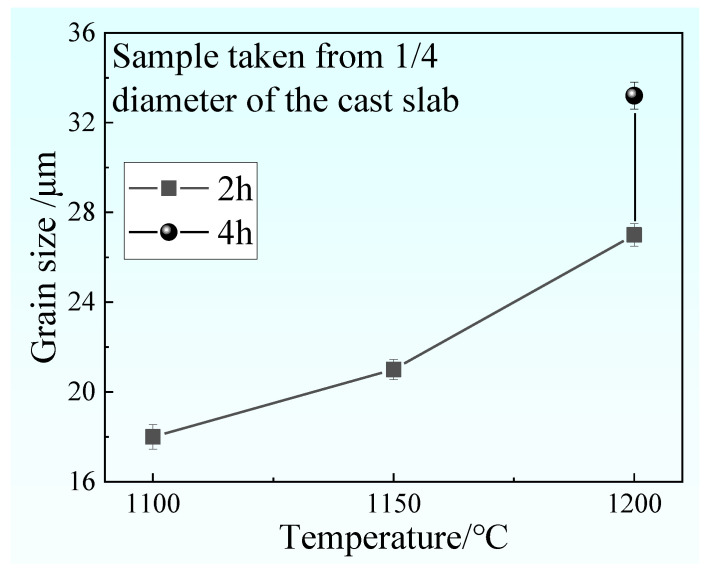
The relationship between austenite grain size and heating temperature and heating time for samples taken from 1/4 diameter of the cast slab.

**Table 1 materials-14-02356-t001:** Chemical composition of 20CrMnTi gear steel used in the study (mass%).

Element	C	Mn	Si	Cr	Ti	P	S	Fe
mass%	0.21	0.995	0.264	1.15	0.0821	0.0132	0.0021	Bal.

## Data Availability

This study did not report any data.

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
