# Peer review of "The Role of Elements Partition and Austenite Grain Size in the Ferrite-Bainite Banding Formation during Hot Rolling"

_materials, 2021, doi:10.3390/ma14092356_

Round 1
Reviewer 1 Report
The article presents the results of a large number of studies and simulation of the behavior of steel (20CrMnTi) during hot rolling. The work was done on a relevant topic. However, there are minor flaws in the article that should be eliminated:
1) The introduction contains a very large number of articles dating from 60 ... 70 years of the last century. It would be nice to provide links (References) to more recent articles.
2) The lines 97... 103 in article are more likely to refer to 2. Materials and methods.
3) Methods and materials do not describe the equipment that was used for rolling, heating and cooling.
4) The simulation results (Fig. 8) should be made with multi-colored lines, since they are poorly distinguishable. Lines 1, 2 and 3 are difficult to distinguish.
Reviewer 2 Report
Dear Authors,
congratulations on the interesting research. I have provided a review with critics and suggestions for your consideration. I believe making the suggested corrections would greatly improve the quality of your paper, which would be beneficial for both you as authors and the journal itself.
Best regards.

Reviewer 3 Report
Thank you for this contribution. This is an interesting and timely manuscript. The conducted analysis is typically standard and falls within the expected work from such a publication and hence the work merits publication. As such, the authors are invited to properly address the following items: 1. How was the heating regime applied based on? As seen in Fig. 1. 2. The details on the Matlab model are quite superficial and light. Please ensure that your model is properly described to enable interested researchers from extending and replicating your work. 3.Author Response
please see the attachment

Round 2
Reviewer 2 Report
Dear Authors,
congratulations on a good job of improving the manuscript based on the Reviewers' suggestions. The manuscript's overall quality has been greatly improved as a result.
Best regards.